# Intensity Modulated Proton Beam Therapy versus Volumetric Modulated Arc Therapy for Patients with Nasopharyngeal Cancer: A Propensity Score-Matched Study

**DOI:** 10.3390/cancers13143555

**Published:** 2021-07-16

**Authors:** Yung-Chih Chou, Kang-Hsing Fan, Chien-Yu Lin, Tsung-Min Hung, Bing-Shen Huang, Kai-Ping Chang, Chung-Jan Kang, Shiang-Fu Huang, Po-Hung Chang, Cheng-Lung Hsu, Hung-Ming Wang, Jason Chia-Hsun Hsieh, Ann-Joy Cheng, Joseph Tung-Chieh Chang

**Affiliations:** 1Proton and Radiation Therapy Center, Department of Radiation Oncology, Linkou Chang Gung Memorial Hospital, Chang Gung University, Taoyuan 333, Taiwan; russellhome88@gmail.com (Y.-C.C.); khs.fan@gmail.com (K.-H.F.); qqvirus1022@gmail.com (C.-Y.L.); min7363@adm.cgmh.org.tw (T.-M.H.); beanson.tw@gmail.com (B.-S.H.); ajchen@mail.cgu.edu.tw (A.-J.C.); 2Department of Radiation Oncology, New Taipei Municipal Tucheng Hospital, New Taipei City 236, Taiwan; 3Department of Otolaryngology-Head Neck Surgery, Linkou Chang Gung Memorial Hospital, Chang Gung University, Taoyuan 333, Taiwan; changkp@adm.cgmh.org.tw (K.-P.C.); handneck@gmail.com (C.-J.K.); bigmac@cgmh.org.tw (S.-F.H.); bc1766@gmail.com (P.-H.C.); 4Division of Medical Oncology, Department of Internal Medicine, Linkou Chang Gung Memorial Hospital, Chang Gung University, Taoyuan 333, Taiwan; hsu2221@adm.cgmh.org.tw (C.-L.H.); whm526@adm.cgmh.org.tw (H.-M.W.); wisdom5000@gmail.com (J.C.-H.H.); 5Department of Medical Biotechnology and Laboratory Science, College of Medicine, Chang Gung University, Taoyuan 333, Taiwan; 6Department of Radiation Oncology, Xiamen Chang Gung Memorial Hospital, Xiamen 361000, China

**Keywords:** IMPT, VMAT, nasopharyngeal cancer

## Abstract

**Simple Summary:**

Based on literature evidence, it is difficult to conclude the advantages and safety of IMPT in patients with NPC. We performed propensity score matching analysis of patients treated with IMPT and VMAT by the same group of physicians within the same institute. Finally, we observed that IMPT reduced the requirement of nasogastric tube insertion and body weight loss during treatment, and the oncologic outcomes were potentially better than that of VMAT. However, IMPT increased the rate of grade III radiation dermatitis. Our current data indicate that IMPT is safe and beneficial as a frontline therapy for patients with NPC.

**Abstract:**

(1) Background: We compared the outcomes of patients with nasopharyngeal carcinoma treated with IMPT and VMAT. (2) Methods: We performed a retrospective propensity score matching analysis (1:1) of patients treated with IMPT (years: 2016–2018) and VMAT (2014–2018). Survival was estimated using the Kaplan–Meier method. Multivariate Cox proportional hazards regression analysis was used to identify the independent predictors of survival. Binary toxicity endpoint analyses were performed using a Cox model and logistic regression. (3) Results: Eighty patients who received IMPT and VMAT were included. The median follow-up time was 24.1 months in the IMPT group. Progression-free survival (PFS) and overall survival (OS) were not statistically different between the two groups but potentially better in IMPT group. In multivariate analysis, advanced N-stage and body weight loss (BWL; >7%) during radiotherapy were associated with decreased PFS. The IMPT group had significantly less requirement for nasogastric (NG) tube placement and BWL during treatment. The mean oral cavity dose was the only predictive factor in stepwise regression analysis, and IMPT required a significantly lower mean dose. However, IMPT increased the grade 3 radiation dermatitis. (4) Conclusions: IMPT is associated with reduced rates of NG tube insertion and BWL through reducing oral mean dose, potentially producing better oncologic outcomes.

## 1. Introduction

Nasopharyngeal carcinoma (NPC) affected an estimated 130,000 patients worldwide in 2018. It is endemic in Southeastern Asia, South China, and North Africa [1]. Radiotherapy (RT) with platinum-based chemotherapy is the current standard treatment [2,3]. Intensity-modulated radiotherapy (IMRT) has significant dosimetric properties compared with 2D or 3D conformal RT, and it increases the local control rates and decreases specific toxicity rates (e.g., xerostomia) [4,5]. However, IMRT increased the scattering dose to the anterior oral cavity and the total integral dose because of the entrance and exit of multiple beam paths [6]. IMRT can cause significant symptoms including mucositis, pharyngitis, dysphagia, xerostomia, nausea, vomiting, and severe body weight loss (BWL). These side effects can limit patients’ compliance with combined modality treatments, increase mortality and morbidity, and irreversibly impair their quality of life (QoL) [7,8]. Volumetric modulated arc therapy (VMAT), an improved iteration of IMRT, is generally considered as the most advanced technology of photon beam therapy; however, its use is limited by similar underlying physical properties of photons [9].

Proton beam therapy is an attractive treatment strategy that decreases unnecessary dose to normal tissues. The inherent physical properties of the Bragg peak deposit maximum radiation dose on the tumor target and eliminate the exit dose beyond the target. The initial technique of “passive scatter” as a form of 3D conformal proton beam is directed to the target using a compensator and aperture. Although passive scatter successfully decreased normal tissue toxicity in different tumors, it has failed to reduce toxicities in patients with NPC, including hearing loss and weight loss, and gastrostomy tube (GT) placement [10]. Multi-field optimization intensity-modulated proton therapy (MFO-IMPT) is a magnetically guided spot scanning proton therapy, in which all proton spots from complex fields are simultaneously optimized by using an inverse treatment planning system [11]. MFO-IMPT can be used to design and deliver high conformal and complex dose distributions to the target, while sparing the organs at risk (OARs). This simultaneous integrated boost treatment plan can be readily created with IMPT, similar to IMRT [12].

Previous studies confirmed the dosimetric advantages between IMPT and IMRT via planning comparisons and demonstrated dose reductions to several OARs without compromising target volume coverage, conformality, and homogeneity [13,14]. While the use of proton beam therapy is rapidly increasing worldwide, limited data are available on clinical outcomes for NPC. To date, only two reports are available with 10 patients in each cohort from a single institution [15,16]. We investigated the outcomes of patients with NPC treated using IMPT from an endemic area, and analyzed whether they showed satisfactory oncologic outcomes and reduced morbidity when compared to propensity score-matched controls treated with VMAT.

## 2. Materials and Methods

### 2.1. Patient Selection

The Institutional Review Board of our hospital approved this study. The study included 80 consecutive patients with histologically proven NPC who received IMPT at the Chang Gung Memorial Hospital between 2016 and 2018. Patients treated with passive scattering proton therapy or a combination of photon therapy and those with a medical history of radiation to the head and neck region were excluded.

Another group comprising 278 patients with NPC treated using VMAT identified between 2014 and 2018 was also included. Patients treated with IMPT were propensity score-matched at a 1:1 ratio with those receiving VMAT based on factors that influenced treatment volumes and expected acute toxicity during RT, with the nearest neighbor method without replacement. The matching criteria were based on N-stage, T-stage, chemotherapy administration, smoking status, sex, age, and Epstein–Barr virus (EBV) status. 

### 2.2. Pre-Treatment Evaluation, Data Collection, and Definition

Pre-treatment evaluation included magnetic resonance imaging (MRI), nasopharyngeal fiberendoscopy, 18-fluorodeoxyglucose-positron emission tomography, EBV DNA quantitative PCR, chest X-ray, and abdominal sonography. Data were collected for baseline patient and tumor characteristics, including age, sex, pathology, staging (based on the eighth edition of AJCC), EBV titer, smoking status (current smoker), presence of comorbidities according to the Charlson comorbidity index, tumor outcomes, emergency room visits, and unplanned hospitalizations. Physicians assessed acute adverse events (AEs) weekly during radiation and the body weight was measured weekly. The incidence and duration of feeding tube utilization were recorded in the medical record. For patients receiving concurrent chemoradiotherapy (CCRT), the physician checked the complete blood count, renal function, and liver function before biweekly CCRT. All AEs were assessed according to the National Cancer Institute Common Terminology Criteria for Adverse Events, version 4.03. 

Overall survival (OS) was defined as the time interval (in months) from the date of first RT to death or time of analysis. Progression-free survival (PFS) is defined as the time between the date of first RT and disease recurrence or death. Patients were censored at their last follow-up date.

### 2.3. Radiotherapy and Chemotherapy

All patients were immobilized using customized thermoplastic masks in the supine position. The oral bite block and mold care pillow were only used for IMPT patients. Computed tomography (CT) simulation with 1.25 mm slice thickness with and without contrast was performed for all patients. MRI simulation was also performed for all patients treated using IMPT. The registration and fusion diagnostic MRI to CT was applied in VMAT patients. Treatment planning calculations were performed using the non-contrast CT scan. 

The clinical target volume and prescribed radiation dose were based on RTOG0225 and prospective clinical trial NRG-HN001 High-risk clinical tumor volume (CTV6996) was defined as a gross disease plus a 3– 5 mm margin, and the prescribed dose was 69.96 Gy for VMAT or 69.96 Gy (relative biological effectiveness (RBE)) for IMPT given in 2.12 Gy or Gy (RBE) fractions. The RBE value of 1.1 was assumed for protons. The high-risk subclinical region (CTV5940) included 59.4 Gy in 33 fractions (1.8 Gy/fraction). A low-risk clinical tumor volume (CTV5412) included 54.12 Gy in 33 fractions (1.64 Gy/fraction) for N0 and/or low neck (level IV and V). For patients treated with VMAT, a 3 or 5 mm planning target volume (PTV) expansion was added to the CTV volumes depending on whether they received image-guided RT. For the PTV-based IMPT optimization plan, the PTV expansion (5–10 mm) was based on setup errors, motion, and range uncertainty. For robust IMPT optimization plan, worst-case robust optimization was used for CTV coverage without PTV expansion. A PTV evaluation (3 mm expansion from CTV volumes) was used for physician to evaluate the treatment plan. 

IMPT plans were generated using the Eclipse planning system (version 13.7; Varian Medical Systems, Palo Alto, CA, USA) with the pencil beam line scanning system. Three different beam angles were used for full-field IMPT plans. There were two different compositions of the three angles: a left and right anterior oblique and a single posterior beam or a left and right posterior oblique and a single rear beam. We used left and right posterior oblique angles for patients with excess dental metal filling because the CT artifacts increased range uncertainty. The planning system optimized all spots from all fields simultaneously. The PTV-based optimization was used initially, and the worst-case robust optimization algorithm was used. The patient-specific quality assurance was measured before treatment delivery. Two-dimensional kilovoltage imaging was performed daily for all patients. The VMAT plan was also generated using the Eclipse planning system with the anisotropic analytical algorithm. Three arcs covered the whole target and optimized treatment objectives simultaneously. Treatment objectives were covering 95% of the PTV with the prescribed dose while minimizing radiation doses to the adjacent OARs for IMPT and VMAT. For all the patients in the study, adaptive re-planning was routinely performed at around the fourth week of treatment.

Most patients received concurrent chemotherapy and no patient received adjuvant chemotherapy. The patients with stage I disease were treated with RT alone. The main concurrent chemotherapy (PUL) regimen was intravenous cisplatin (P) (50 mg/m^2^, day 1) and oral tegafur plus uracil (U) (300 mg/m^2^/day) plus leucovorin (L) (60 mg/day) daily for 14 days (DeCesaris et al., 2019). The other concurrent chemotherapy regimen was intravenous cisplatin 40 mg/m^2^ weekly for 6–7 cycles. Less patients with advanced disease received induction chemotherapy. The main induction chemotherapy (GP) regimen was gemcitabine (1000 mg/m^2^, day 1, day 8) and cisplatin (60–75 mg/m^2^, day 1) once every 3 weeks for 3 cycles. The second most common induction chemotherapy (TP) regimen was docetaxel (60–75 mg/m^2^, day 1) and cisplatin (60–75 mg/m^2^, day 1) once every 3–4 weeks for 3 cycles.

### 2.4. Statistical Analysis 

After propensity score matching, the intergroup differences in categorical variables were compared using the chi-squared test. The numerical variables were compared using continuous variables tested using independent Student’s t-tests. The composite endpoint of nasogastric (NG) tube insertion or BWL (>7%) served as the primary endpoint. Multivariate logistic regression with age dichotomized at 60 years as a covariate was used to provide an odds ratio (OR) for predictors of the primary endpoint for all variables simultaneously. OS and PFS were estimated using the Kaplan– Meier method. Multivariate Cox proportional hazards regression analyses were used to identify the independent predictors of PFS. All tests were two-sided and a *p*-value < 0.05 was considered significant. Statistical analyses were conducted using the statistical software package SPSS version 25.0 plugin with the PSMATCHING3.04.spe of the R program, version 3.3.0 (Vienna, Austria).

## 3. Results

### 3.1. Patient and Tumor Characteristics

Patient and treatment characteristics are summarized in Table 1. There were no imbalances between the two groups in any covariates. Eight percent of the patients were males. The median age was 47.6 years (22.6–79.2) and 50.1 years (27.3–79.2) in the IMPT and VMAT groups, respectively. A majority of the patients showed good performance status with the Charlson comorbidity index of 0 to 1 in both groups at NPC diagnosis. EBV PCR titer (>200 copies/mL) at diagnosis was observed in 60% of patients. Approximately 10% of patients treated using RT alone had stage I disease. Ninety percent of patients had stage II–IV disease and approximately 12.5% received induction chemotherapy followed by concurrent chemotherapy. All patients in the two groups completed radiotherapy as planned. The details of dose coverage and conformity index between two groups were listed in Appendix A. The IGRT was performing in 92.5% and 100% of patients in VMAT and IMPT groups, respectively. The induction chemotherapy was completed in the two groups. In total, 64 (92.8%) of 69 in the IMPT group and 58 (84.1%) of 69 in the VMAT group achieved a high dose of cisplatin (≥200 mg/m^2^) during CCRT.

### 3.2. Oncological Outcome

Median follow-up time was 24.1 months (18.2–34.3) and 42.2 months (18.1–62.6) for patients treated using IMPT and VMAT, respectively. Nine patient deaths were recorded in the VMAT group, whereas no patient died in the IMPT group. The two-year OS rates were 100% and 89.5% for the IMPT and VMAT groups, respectively (Figure 1a). Twenty-two events (recurrence or death) were observed, four in the IMPT group and eighteen in the VMAT group, with a two-year PFS rate of 94.4% and 83.7% in the IMPT and VMAT groups, respectively (Figure 1b). The univariate and multivariate analyses for PFS are presented in Table 2, advanced N-stage (hazard ratio (HR) = 6.912; 95% confidence interval (CI): 1.877–25.456; *p*-value = 0.004) and BWL (>7%) during RT (HR = 3.216; 95% CI: 1.062–9.742; *p*-value = 0.039) were associated with a decreased PFS. The Kaplan– Meier curve analysis based on BWL (>7%) during RT is shown in Appendix A. The HR between IMPT and VMAT in multivariate analysis was 0.513 (95% CI: 0.12–2.5, *p*-value = 0.436). Overall, there were five locoregional relapses, two in the IMPT group and four in the VMAT group. Sixteen distant relapses were observed, two in the IMPT group and 14 in the VMAT group. 

### 3.3. NG Tube Placement, BWL, and Radiation Dose Difference

Toxicity endpoints between treatment groups are described in Table 3. Four patients (5%) treated with IMPT required NG tube placement compared to 12 (15%) treated with VMAT (*p*-value = 0.026). The mean duration of NG tube placement was 3.8 and 7.4 weeks in the IMPT and VMAT groups, respectively. The mean percentage of BWL during RT was 4.87% in the IMPT group and 6.21% in the VMAT group (*p*-value = 0.038). Twenty-four patients (32.4%) treated with IMPT and 41 patients (54.7%) treated with VMAT had BWL (>7%) (*p*-value = 0.006). When considering NG tube placement or BWL (>7%) as endpoints, OR for the radiation modalities was 0.358 (95% CI: 0.188–0.680, *p*-value = 0.002) in univariate logistic regression (Table 4); using multivariate logistic regression analysis, OR for the radiation modality and T-stage were 0.302 (95% CI: 0.150–0.607, *p*-value = 0.001) and 2.195 (95% CI: 1.072–4.493, *p*-value = 0.031), respectively (Table 4). Patients administered with IMPT received significantly reduced mean doses to the oral cavity, superior constrictor muscle, middle constrictor muscle, and inferior constrictor muscle (Appendix A). The representative figures regarding IMPT and VMAT treatment planning dose are in Appendix A. When radiation dose was added as a covariate in the multivariate analysis, the mean oral dose, instead of radiation modality and T stage, was significantly associated with NG tube placement or weight loss (>7%) (Table 4).

### 3.4. Common Acute AEs

RD on the neck was the most common AE observed, and grade 3 RD, which required wound care, was a significant AE between the IMPT [*n* = 28 (35%)] and VMAT [*n* = 6 (7.5%); (*p* < 0.000)] groups (Table 2). Severe dermatitis usually occurred at the 5th–6th week after RT and lasted approximately 5–6 weeks with skin care. There were no significant differences in acute grade 3 mucositis and grade 2–3 xerostomia during RT between the IMPT and VMAT groups. No differences were observed between the two groups with respect to the frequency of emergency room visits or unscheduled hospitalizations.

## 4. Discussion

This study confirmed several significant findings: patients with NPC treated with MFO-IMPT showed reduction of treatment-related toxicities, compliance with combined chemotherapy and RT, and excellent oncologic outcomes. Although MFO-IMPT is an effective treatment plan and can precisely deliver dose to target voxel-by-voxel, proton dosimetry is highly sensitive to target depth and tissue heterogeneity. Daily fluctuations in patient position, changes in anatomy owing to tumor regression or weight loss, and the presence of image artifacts impede the accuracy of treatment delivery [12]. Special efforts were undertaken to reduce these risk factors such as precision in patient setup with image guidance, re-planning for every patient during treatment, and choosing the right beam paths to avoid image artifacts and possibly the path to the sinusitis area, as the beam depth will change during RT. The treatment planning system commissure and patient specific quality assurance were also required to ensure the treatment fidelity and integrity. With a two-year follow-up of eighty patients, this study demonstrated that IMPT could be safely administered in patients with NPC as a frontline therapy instead of IMRT. 

IMPT significantly reduced the rates of NG tube placement and the mean percentage of BWL. Our study is consistent with a previous case study, which showed that IMPT decreased feeding tube placement rates in patients with NPC [16]. However, a 65% rate of feeding tube usage in IMRT was reported, which is more than that reported in previous studies (i.e., 20–30% in grade 3–4 mucositis) [17,18,19]. In our study, occurrence of grade 3 mucositis (17.5%) was consistent with previous reports on IMRT, and we demonstrated that the NG tube placement rate reduced from 15% in VMAT to 7.8% in IMPT. BWL (5–8%) and dehydration were the most common reasons for NG tube insertion in this study. However, the final decision on NG tube placement was made after a discussion between the patient and the physician. When patients refused NG tube placement, it resulted in decreased body weight. Both feeding tube placement and BWL resulted from insufficient intake during RT; therefore, we combined these two factors as a composite endpoint. Our study found that a decreased mean dose to the oral cavity was the reason for a reduction in feeding tube placement and BWL in patients receiving IMPT. The mean radiation dose to the oral cavity was the only independent predictor in our composite endpoint analysis, and it was significantly different between the IMPT (18 GyE) and VMAT (38 Gy) groups, which was similar to previous studies [14,15,16]. With the Bragg peak of proton beam, mucositis occurs mainly in posterior parts of the oral cavity or oropharynx. Enough oral intake through the intact oral mucosa could maintain body weight and reduce the requirement for a feeding tube.

By analyzing the prognostic factors in the entire cohort, we observed that BWL (>7%) and N-stage were the only independent prognostic factors for PFS. While considering survival and oncologic outcomes, the feeding tube placement and BWL were categorized as two variables. Because these variables represented different means in clinical practice, feeding tube order is an intervention maintaining nutritional intake and possibly improving outcomes [20]. In contrast, BWL is a malnutrition status, which correlated with poor prognosis in patients with NPC [21,22]. The underlying reason mainly involves malnutrition, which correlated with short-term mortality, immune dysfunction, and treatment interruption in patients with head and neck cancer [23,24]. However, significantly longer radiation time, poor chemotherapy compliance, and reduction in neutrophil count could influence treatment outcomes [25,26,27].

By reducing toxicity during treatment as mentioned, the IMPT group might potentially improve overall survival and disease control outcomes compared to the VMAT group. In our study, only one local recurrence, one regional recurrence, and two cases of distant metastasis were noted among eighty patients after two-year follow-up in the IMPT group. Lewis et al. also reported 100% local control rate and one patient had distant metastasis among ten patients with NPC [15]. In the study, although there was no statistical significance in OS or PFS between IMPT and VMAT, the *p*-values obtained for OS and PFS were 0.099 and 0.071, respectively. Larger sample size or more follow-up time may further impact and observe statistically significant results.

RD was an important acute side effect in the IMPT group in our study—grade 3 RD: 35% (IMPT group) and 7.5% (VMAT group). Lewis et al. also revealed that 40% of patients receiving IMPT had grade 3 RD [15]. In a physical setting, the megavoltage photon beam, as an indirect ionizing radiation, builds up the radiation dose by depth, and spares the skin that results in reduced RD; however, the proton beam, as a direct ionizing radiation, deposits radiation dose on the skin at the entrance. Similarly, the occurrence of RD has been related to proton therapy in breast cancer treatment. DeCesaris et al. recently demonstrated that proton radiation significantly increased grade 2 RD rate compared with photon therapy [28]. The degree of RD was more severe in NPC because most of the patients required a high dose (70 Gy) to the gross metastatic lymph node; moreover, gross lymph node regression produced unexpected hot spot dose on the skin. RD caused physical, emotional, and functional discomfort that impaired patients’ QoL, with pronounced effects in high-grade dermatitis [29]. Although all RD healed after intensive interventions in this study, further proton therapy studies should focus on reducing RD [30].

To the best of our knowledge, this study presented the largest cohort of patients with NPC treated using IMPT in an endemic area. Furthermore, clinical outcomes and toxicity were compared with propensity score-matched patients treated using VMAT by the same group of physicians within the same institution. Therefore, this study design could serve as an internal control. However, there were several study limitations. Given the retrospective nature of the analysis, the subjective toxicity might be underestimated; therefore, feeding tube insertion and weight loss were the main endpoints in the study instead of mucositis and dysphagia. Secondly, several crucial long-term toxicities, a health-related QoL study, and patient-reported outcomes are still being followed-up and are in preparation. Thirdly, VMAT is covered by National Health Insurance (NHI) in our country, while IMPT is not covered by NHI. This may lead to potential selection bias regarding socioeconomic status between the two groups.

## 5. Conclusions

This propensity score matching analysis of patients with NPC treated using either IMPT or VMAT suggested that IMPT could significantly reduce the need for feeding tube insertion and BWL by decreasing the mean dose to the oral cavity with potential benefits for tumor control. Our current data indicate that IMPT is safe and beneficial as a frontline therapy for patients with NPC.

## Figures and Tables

**Figure 1 cancers-13-03555-f001:**
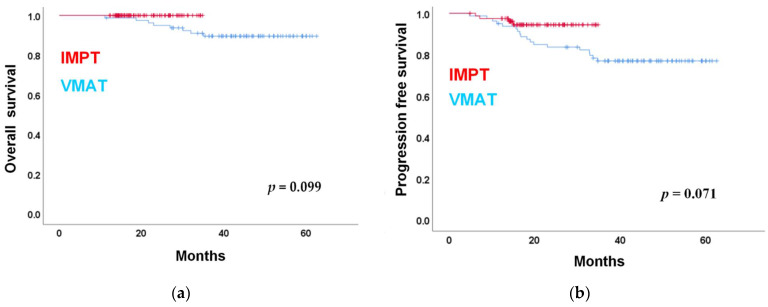
(**a**) Overall survival; (**b**) progression-free survival. Kaplan–Meier analysis of the IMPT and VMAT patients. Abbreviations: IMPT: Intensity-modulated proton therapy, VMAT: Volumetric modulated arc therapy.

**Table 1 cancers-13-03555-t001:** Comparison between characteristics of patients with NPC treated using IMPT and VMAT.

Characteristics	IMPT (*n* = 80)	VMAT (*n* = 80)	*p*-Value
Age at diagnosis, mean (IQR), y	47.6 (22.6–79.2)	50.1 (27.3–79.2)	
Age, *n* (%)			0.844
>60 y/o	65 (81.3)	63 (78.8)	
<60 y/o	15(18.7)	17 (21.2)	
Sex, *n* (%)			0.284
Male	64 (80)	70 (87.5)	
Female	16 (20)	10 (12.5)	
Charlson Comorbidity Index			0.298
0–1	69 (86.3)	63 (78.8)	
≥2	11 (13.8)	17 (21.3)	
WHO type, *n* (%)			0.845
I	1 (1.3)	1 (1.3)	
II	24 (30.0)	21 (26.3)	
III	55 (68.8)	58 (72.5)	
T-stage, *n* (%)			0.760
T1	32 (40.0)	25 (43.8)	
T2	11 (13.8)	8 (12.5)	
T3	19 (23.8)	7 (17.5)	
T4	18 (22.5)	14 (26.3)	
N-stage, *n* (%)			0.909
N0	13 (16.3)	10 (20.0)	
N1	39 (48.8)	34 (46.3)	
N2	15 (18.8)	12 (16.3)	
N3	13 (16.3)	18 (17.5)	
AJCC 8th stage, *n* (%)			0.945
I	8 (10.0)	9 (11.3)	
II	21 (26.3)	21 (26.3)	
III	19 (23.8)	16 (20.0)	
IV	32 (40.0)	34 (42.6)	
EBV PCR titer, *n* (%)			0.257
>200	45 (56.3)	52 (65.0)	
<200	35 (43.8)	28 (35.0)	
Treatment modality, *n* (%)			1.000
Chemo with CCRT	10 (12.5)	10 (12.5)	
CCRT	59 (73.8)	59 (73.8)	
RT alone	11(13.7)	11(13.7)	
Induction chemotherapy			0.531
GP	9 (90.0)	8 (80.0)	
TP	1 (10.0)	2 (20.0)	
Concurrent chemotherapy regimen			0.459
PUL	58 (84.1)	61 (88.4)	
Weekly cisplatin	11 (15.9)	8 (11.6)	
Cisplatin total dose (mg/m^2^)			0.111
<200	5 (7.2)	11 (15.9)	
≥200	64 (92.8)	58 (84.1)	
Smoking at diagnosis, *n* (%)			0.817
No	43 (54.4)	45 (56.3)	
Yes	36 (45.6)	35 (43.8)	

Abbreviations: IMPT—Intensity-modulated proton therapy, VMAT—Volumetric modulated arc therapy, IQR—Interquartile range, EBV—Epstein–Barr virus, CCRT—Concurrent chemoradiotherapy, WHO—World Health Organization, AJCC—American Joint Committee on Cancer, PCR—Polymerase chain reaction, PUL—Cisplatin (P) and tegafur plus uracil (U) plus leucovorin, GP—Gemcitabine and cisplatin, TP—Docetaxel and cisplatin.

**Table 2 cancers-13-03555-t002:** Univariate and multivariate analysis for progression-free survival.

Characteristics	Univariate Analysis			Multivariate Analysis		
	OR	95% CI	*p* Value	OR	95% CI	*p* Value
Age						
<60	1					
≥60	1.618	0.532–4.920	0.396			
Sex						
female	1			1		
male	0.852	0.195–3.717	0.831	1.618	0.321–8.155	0.560
Pathology						
WHO I, II	1			1		
WHO III	1.023	0.363–2.879	0.966	1.257	0.381–4.143	0.707
T stage						
I–II	1			1		
III–IV	1.358	0.539–3.425	0.516	1.267	0.437–3.672	0.663
N stage						
0–I	1			1		
II–III	3.995	1.539–10.374	0.004 *	6.912	1.877–25.456	0.004 *
Radiation modality						
VMAT	1			1		
IMPT	0.372	0.122–1.133	0.082	0.298	0.088–1.011	0.052
EBV titer						
≥200	1			1		
<200	0.447	0.059–3.362	0.434	0.326	0.075–1.409	0.133
Induction chemotherapy						
No	1			1		
Yes	2.679	0.952–7.541	0.062	2.235	0.669–7.466	0.191
Smoking at diagnosis						
No	1			1		
Yes	1.933	0.749–4.988	0.173	2.484	0.886–6.912	0.084
Weight loss ≥7%						
No	1			1		
Yes	2.952	1.052–8.283	0.040 *	3.216	1.062–9.742	0.039 *
Nasogastric tube insertion						
No	1			1		
Yes	0.631	0.144–2.759	0.541	0.566	0.121–2.644	0.470
Charlson Comorbidity Index						
0–1	1			1		
≥2	1.383	0.455–4.205	0.567	1.927	0.601–6.180	0.270
Cisplatin total dose (mg/m^2^)						
<200	1			1		
≥200	0.367	0.049–2743	0.328	0.318	0.038–2.660	0.290

Abbreviations: OR—Odds ratio, CI—Confidence interval, IMPT—Intensity-modulated proton therapy, VMAT—Volumetric modulated arc therapy, EBV—Epstein–Barr virus, WHO—World Health Organization; * means statistically significant.

**Table 3 cancers-13-03555-t003:** Toxicity Analysis for planned endpoints between IMPT and VMAT plans.

Variables	IMPT (*n* = 80)	VMAT (*n* = 80)	*p*-Value
NG tube placement, No. (%)	4 (5.0)	12(15.0)	0.026
Percentage Body weight loss (SD)	4.87 (3.94)	6.21 (4.15)	0.038
Weight loss over 7%, No. (%)	24 (32.4)	41 (54.7)	0.006
Weight loss over 7% or NG tube during treatment, No. (%)	25 (32.9)	46 (57.5)	0.002
Grade 3 dermatitis with wound care, No. (%)	28 (35)	6 (7.5)	<0.000
Grade 3 mucositis, No. (%)	8 (10.0)	14 (17.5)	0.178
Grade 2–4 Xerostomia, No. (%)	9 (11.3)	13 (16.3)	0.358
Emergency Room Visit	7 (8.8)	13 (16.3)	0.151
Unscheduled Hospitalization	8 (10.0)	7 (8.8)	0.786

Abbreviations: IMPT—Intensity-modulated proton therapy, VMAT—Volumetric modulated arc therapy, NG—Nasogastric.

**Table 4 cancers-13-03555-t004:** Univariate and multivariate analysis of the association with weight loss (≥7%) or nasogastric tube insertion.

Characteristics	Univariate Analysis			MVA without Radiation Dose			MVA with Radiation Dose		
	OR	95% CI	*p-*Value	OR	95% CI	*p-*Value	OR	95% CI	*p-*Value
Age									
<60	1			1			1		
≥60	0.562	0.234–1.350	0.197	0.419	0.156–1.126	0.085	0.445	0.167–1.183	0.105
Sex									
female	1			1			1		
Male	0.995	0.429–2.311	0.991	1.110	0.455–2.712	0.818	1.168	0.468–2.913	0.739
Pathology									
WHO I, II	1			1			1		
WHO III	0.858	0.434–1.697	0.660	0.864	0.406–1.839	0.704	0.956	0.439–2.081	0.909
T stage									
I–II	1			1			1		
III–IV	1.985	1.055–3.737	0.034 *	2.195	1.072–4.493	0.031 *	1.827	0.865–3.858	0.114
N stage									
0–I	1			1			1		
II–III	1.487	0.772–2.864	0.235	1.742	0.751–4.042	0.196	1.429	0.590–3.460	0.429
Radiationmodality									
VMAT	1			1			1		
IMPT	0.358	0.188–0.680	0.002 *	0.302	0.150–0.607	0.001 *	0.828	0.238–2.888	0.768
Mean oralcavity dose	1.063	1.031–1.095	0.000 *				1.069	1.003–1.140	0.038 *
Mean superior Constrictor muscle dose	0.997	0.989–1.005	0.441				0.994	0.977–1.011	0.472
Mean middle Constrictor muscle dose	1.035	1.002–1.069	0.036 *				0.984	0.946–1.023	0.407
Mean inferior Constrictor muscle dose	1.043	1.001–1.085	0.042 *				1.010	0.980–1040	0.531
EBV titer									
≥200	1			1			1		
<200	1.729	0.906–3.298	0.097	1.024	0.438–2.394	0.957	1.186	0.481–2.922	0.711
Chemotherapy modality									
No	1			1			1		
Concurrent	2.373	0.854–6.591	0.097	1.691	0.521–5.487	0.382	1.368	0.412–4.542	0.608
Induction	1.436	0.391–5.269	0.585	0.759	0.165–3.497	0.724	0.612	0.129–2.908	0.537
Smoking									
No	1			1			1		
Yes	1.017	0.544–1.900	0.959	0.981	0.489–1.970	0.958	1.094	0.526–2.275	0.811

Abbreviations: MVA—Multivariate analysis, OR—Odds ratio, CI—Confidence interval, IMPT—Intensity-modulated proton therapy, VMAT—Volumetric modulated arc therapy, WHO—World Health Organization; * means statistically significant.

## Data Availability

The data presented in this study are not publicly available for privacy and legal reasons.

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
