# Peer review of "Intensity Modulated Proton Beam Therapy versus Volumetric Modulated Arc Therapy for Patients with Nasopharyngeal Cancer: A Propensity Score-Matched Study"

_cancers, 2021, doi:10.3390/cancers13143555_

Round 1

Reviewer 1 Report

In this paper, many cases with nasopharyngeal carcinoma are discussed.

Please continue to increase the number of cases and consider in more detail.

Author Response

[Reviewer 1 general comment]

In this paper, many cases with nasopharyngeal carcinoma are discussed.

Please continue to increase the number of cases and consider in more detail.

[Response]: Thank you for your comment. We will continue to extend patient data and analysis in our further research.

Reviewer 2 Report

The munuscript is of high quality, interesting, well-structured, clear, and with usefull results 

I have only one comment:

The induction/concomitant chemotherapy is an important factor influencing acute toxicity but only one regimen (the main concomitant regimen) is marginally mentioned. Furthermore, the referrence number (28) is wrong, I am afraid.  I recommend to extend the information about induction and concomitant therapy and regimens used in the study. 

Author Response

Reviewer 2:

[Reviewer 2 general comment ]

The manuscript is of high quality, interesting, well-structured, clear, and with usefull results.

[Response]: We greatly appreciate the reviewer’s supportive comments on our

manuscript.

[Reviewer 2 comment ]

The induction/concomitant chemotherapy is an important factor influencing acute toxicity but only one regimen (the main concomitant regimen) is marginally mentioned. Furthermore, the reference number (28) is wrong, I am afraid. I recommend to extend the information about induction and concomitant therapy and regimens used in the study. 

[Response]: We appreciate your comment on the lack of chemotherapy regimen detail we used. The two concurrent chemotherapy regimens and two induction chemotherapy regimens were added in the Material and method 2.3 chemotherapy (p. 4). The difference in chemotherapy used between IMPT and VMAT patients was listed in results Table 1. The induction chemotherapy was smoothly completed in both groups. There was no difference between the two groups in the achievement of a high dose of cisplatin (≥200 mg/m2 ) during CCRT(Result 2.1 ).

Reference number 28 is correct but the first author's name was wrong in the manuscript. This has now been amended.

Reviewer 3 Report

Thank you for the opportunity to review this paper. One can find an increasing interest in the field of proton therapy, particularly IMPT. This article is based on the evaluation of 80 consecutive patients treated with IMPT. This is a hot topic, however, I have some doubts about methodology.

-There is a discrepancy between numbers of patients treated annually in 2016 and 2018 (80 pts) vs. 2014-2018 (278). Resons should be comment.

- Many factors that potentionaly influenced treatment volumes were used for matching. Why GTV was not used stratification?

- IMPT is presented with precise efforts like MR simulation, IGRT, adaptation in fourth week. What about VMAT?

- CTV-PTV expansion seems to be higher for IMPT. Were PTVs for IMPT and VMAT different?

- more dosimetric parameters like conformity index, coverage, etc. should be added

- As follow up time differ between VMAT and IMPT the numbers of deaths as well as relapses are not relevant

- As long period was evaluated was the preferred chemotherapy regime different from 2014 to 2018?

- Similarly, were the strategy for prophylactic NG tube insertion and nutritional support changing during years?

In discussion section there are imprecise sentences, e.g.:

MFO-IMPT showed reduction of toxicities...But skin toxicity was higher for IMPT. Actually, how was skin sparing performed for IMPT? PTV with inner margin 3 mm? Or different way?

Prospective multicentre trial may be another good method? This is different level of evidence!

Why were selected aspects of toxicity evaluated only? What about temporal lobe necrosis for example?

Author Response

Reviewer 3:

[Reviewer 3 general comment]

Thank you for the opportunity to review this paper. One can find an increasing interest in the field of proton therapy, particularly IMPT. This article is based on the evaluation of 80 consecutive patients treated with IMPT. This is a hot topic, however, I have some doubts about methodology.

[Response]: We greatly appreciate the reviewer’s valuable contribution.

[Reviewer 3 comment 1]

There is a discrepancy between numbers of patients treated annually in 2016 and 2018 (80 pts) vs. 2014-2018 (278). Reasons should be comment.

[Response]: This is a very interesting point to clarify. In fact, the number of nasopharyngeal cancer patients treated each year in the past ten years is almost the same, with approximately 100-150 patients each year. The main reason why only 80 patients received IMPT from 2016 to 2018 was that VMAT is covered by the National Health Insurance (NHI) in my country, and proton therapy is an expensive technology that NHI does not cover. Most of the patients still received VMAT from 2016 to 2018. The potential socioeconomic selection bias was added to the discussion (Page.10).

[Reviewer 3 comment 2]

Many factors that potentially influenced treatment volumes were used for matching. Why GTV was not used stratification?

[Response]: We believe the gross tumor volume was highly correlated with T and N stages in the AJCC staging system. The T and N stages influenced the radiation treatment volumes and provided segregation of survival outcomes. Therefore, T and N stages were mainly matched

[Reviewer 3 comment 3]

IMPT is presented with precise efforts like MR simulation, IGRT, adaptation in fourth week. What about VMAT?

[Response]: Thank you for this question. In the VMAT group, the registration and fusion diagnostic MRI to CT is applied for GTV delineation. The IGRT was also performed in almost every patient in the VMAT group. The adaptive re-planning was routinely performed during around fourth week of treatment. The information was added on pages 3 and 4.

[Reviewer 3 comment 4]

CTV-PTV expansion seems to be higher for IMPT. Were PTVs for IMPT and VMAT different?

[Response]: PTV expansion for proton therapy included setup errors (3 mm) and 2% range uncertainty. Therefore, the PTV for IMPT was definitely larger than PTV for VMAT. In the future, we tried to analysis whether the setup errors could be reduced because of the precision in patient setup and relative well local-regional control. However, a PTV-evaluation (3 mm expansion from CTV volumes) was used for physician to evaluate the treatment plan, just like VMAT. (Page 3)

[Reviewer 3 comment 5]

More dosimetric parameters like conformity index, coverage, etc. should be added

[Response]: Thank you for the great suggestion. The target dose coverage and conformity index (CI) were listed in the new supplement 1. The PTV6996 V100% coverage was worse in the IMPT because the advanced nasopharyngeal cancer usually closed to the critical organ like the brain stem, spinal cord, optic nerve, and optic chiasm. The proton beam delivered from only three different beam angles reduced the capability of critical organ sparing, which means more PTV coverage was sacrificed for the critical organ sparing. Fortunately, the local control was still great in this study. The PTV5940 coverage was similar between IMPT and VMAT, but the PTV5940 conformity index was higher in the IMPT group. This also evidenced that proton therapy had less conformity in the high dose target volume. In fact, the benefit of proton therapy is sparing normal organs from the low to intermediate radiation dose.

Reviewer 3 comment 6]

As follow up time differ between VMAT and IMPT the numbers of deaths as well as relapses are not relevant.

[Response]: Thank you for your valued comment. Most of the local-regional relapses happened in the first two years after treatment in patients with NPC. The median follow-up time was 24.1 months (18.2–34.3) in IMPT patients, which is relatively enough for local-regional control comparison with VMAT. The goal of PFS comparison was to evidence that the tumor control was not sacrificed for reduction of radiation toxicity. The goal of comparison of overall survival between IMPT and VMAT patients was to reveal the safety of IMPT.

The patients who had loco-regional recurrence or metastasis will receive further aggressive chemotherapy or immunotherapy or target therapy. The salvage rate is relatively good. That may explain why the correlation between death and relapse are not closely relevant. We will present the data in the future study.

Reviewer 3 comment 7]

As long period was evaluated was the preferred chemotherapy regimen different from 2014 to 2018?

[Response]: The chemotherapy treatment guideline was not changed during this period.

Reviewer 3 comment 8]

Similarly, were the strategy for prophylactic NG tube insertion and nutritional support changing during years?

[Response]: The strategy of nutritional support and NG tube insertion did not bed changed during the years.

Reviewer 3 comment 9]

MFO-IMPT showed reduction of toxicities...But skin toxicity was higher for IMPT. Actually, how was skin sparing performed for IMPT? PTV with inner margin 3 mm? Or different way?

[Response]: For VMAT, we did PTV with 3mm away from the body surface during radiation dose planning for reducing radiation dermatitis. However, the method did not use in proton therapy. First, the PTV is created by calculation of setup error and range uncertainty in IMPT. The PTV could even exceed the body surface after computer calculation. Second, the entrance dose of proton beams (70%) at the skin were impossible to avoid because there were only three different beam angles. Third, the proton did not have skin-sparing properties in physics.

[Reviewer 3 comment 10]

Prospective multicenter trial may be another good method? This is different level of evidence!

[Response]: Thank you for highlighting this, we apologize for the imprecise writing and it is easy to misunderstand. The sentence“However, this is still hypothesis generation. Prospective multi-center clinical trials may be another good method to confirm the findings” has been removed. (Page 9)

[Reviewer 3 comment 11]

Why were selected aspects of toxicity evaluated only? What about temporal lobe necrosis for example?

[Response]: Thank you for this suggestion. The median follow-up time was 24.1 months in IMPT patients. The long-term toxicities data could not collect accurately right now. For example, the median latency of temporal lobe necrosis development was around 36 months [3]. Therefore, we were focused on the acute toxicity analysis in the study. Given the retrospective nature of the analysis, the subjective toxicity might be underestimated; we preferred objective feeding tube insertion and weight loss as endpoints. We will continue to follow up on the long-term toxicities and provided the data in our further research.

Zhang, Y.Y., et al., Brain-Specific Relative Biological Effectiveness of Protons Based on Long-term Outcome of Patients With Nasopharyngeal Carcinoma. Int J Radiat Oncol Biol Phys, 2021. 110(4): p. 984-992.